# Recovery of Antimicrobials and Bioaccessible Isoflavones and Phenolics from Soybean (*Glycine max*) Meal by Aqueous Extraction

**DOI:** 10.3390/molecules24010074

**Published:** 2018-12-26

**Authors:** Cyntia S. Freitas, Genilton Alves da Silva, Daniel Perrone, Mauricio A. Vericimo, Diego dos S. Baião, Patrícia R. Pereira, Vânia M. F. Paschoalin, Eduardo M. Del Aguila

**Affiliations:** 1Instituto de Química, Universidade Federal do Rio de Janeiro, Avenida Athos da Silveira Ramos 149, 21941-909 Rio de Janeiro, Brazil; cyntia.freitas@yahoo.com.br (C.S.F.); geniltonalves@gmail.com (G.A.d.S.); danielperrone@iq.ufrj.br (D.P.); diegobaiao20@hotmail.com (D.d.S.B.); biopatbr@gmail.com (P.R.P.); emda@iq.ufrj.br (E.M.D.A.); 2Instituto de Biologia, Universidade Federal Fluminense, Niterói, 4020-141 Rio de Janeiro, Brazil; vericimo@vm.uff.br

**Keywords:** extruded-soy material, polyphenols, isoflavones, bioaccessibility, non-cytotoxic, antimicrobial and antimitogenic activities

## Abstract

Soybeans display strategic potential in food security as a source of protein and functional bioactives for human consumption. Polyphenols and other bioactive compounds can be recovered after an aqueous extraction from soybean meal, a byproduct of soy oil refining. The objective of the present study was to compile and quantify compounds from soybean oil refinery by-products, providing information about valuable bioactive phytochemicals, their bioaccessibility and potential bioactivities. Genistin, daidzin, glycitin and malonylgenistin were the predominant isoflavones, and the overall bioaccessibility of their glycosidic forms was of nearly 75%. Sixteen phenolics were identified and caffeic acid, 5-caffeoylquinic chlorogenic acid and hesperidin were the most predominant. Approximately 30% of gallic acid, syringic acid, vanillic acid and myricetin were released and the antioxidant capacity of aqueous extract was enhanced after simulated in vitro gastro intestinal digestion. The ability of aqueous soybean meal extract to inhibit lipid peroxidation was higher than natural and synthetic food antioxidants. Antimicrobial activity against several foodborne pathogens and antitumoral activity towards human glioblastoma cell line were also observed, but the aqueous extract showed no cytotoxicity to healthy murine cells. Compounds derived from the aqueous soybean meal extract have the potential to be used as health promoting agents.

## 1. Introduction

Over the years, plant extracts have been demonstrated to be a rich source of bioactive compounds able to retard degradation, improve the quality and increase the nutritive value of foods and even provide health benefits [1]. 

Soybean (*Glycine max*), a leguminous plant, is a resource of several phytochemicals and the second largest source of vegetable oil worldwide (after palm oil) with a global production of 271 million metric tons per annum [2] and a high economical value in both national and international markets. The United States of America, Brazil and Argentina are the largest producers of soybeans, reaching together approximately 80% of the cropped grains [3]. 

The large interest in the cultivation of soybean relies on their nutritional value and the health benefits promoted by their components. In addition to its content in protein and lipids, the soybean is a natural resource of phenolic compounds and isoflavones [4]. This crop is rarely consumed by cooking the grain and traditionally it is processed to originate soybean-based food or ingredients for the food industry. To extract oil, soybean grains are mechanically pressed or treated with organic solvents (usually hexane) in order to generate a solid by-product at the end of the process, named soybean meal that still retains most of the protein content found in the grains. For this reason, soybean meal is not discarded but mostly used to produce animal feed, which requires a prior heat treatment followed by meal toasting to eliminate anti-nutritional factors, especially trypsin inhibitors and lectins [4,5]. 

It has been reported that soybean meal can still retain functional bioactive compounds, such as antimicrobial peptides and polyphenols, which could aggregate value to this by-product, stimulating its use as a source of health promoting agents [6]. However, the aforementioned soybean oil extraction requires very expensive equipment to install and maintain and can lead to undesirable side effects, such as impairment of quality and sensorial oil characteristics. In addition, thermal degradation of many beneficial bioactive compounds may occur. 

Xiao et al. [7] and Flachowsky et al. [8] have reported that defatted soybean meal could be an isoflavone source, identifying eight distinct compounds following methanol extraction.

Polyphenols or phenolic compounds are accumulated in soya as products of the secondary metabolism synthesized by attempts throughout their normal development, for a stress response, such as infections, UV radiation or injury. They contribute to the protection against predators and pathogens and to the organoleptic characteristics, conferring flavor and sensory properties such as sweetness, bitterness and astringency. Soybeans may contain isoflavones, anthocyanins, fungal acids, saponins, phenolic acids, hydroxybenzoic and hydroxycinnamic acids, isoflavonoids and isoflavones. Soybean isoflavone content ranges from 0.44 to 5.24 mg/g [9,10,11], which is distributed in three chemical forms, including aglycones and their β-glucoside conjugates: glucosides, malonyl glucosides and acetyl glucosides [12,13,14,15].

Isoflavones possess high antioxidant activity and metal-ion chelating properties, but their major importance is as phytoestrogens, binding to both α and β-estrogen receptors, and causing an estrogenic or anti-estrogenic effect according to the type of estrogen receptor. Their intake has been associated with a decreased risk of hormone-related cancers [11,12]. Furthermore, soya phytochemicals can display several biological activities such as antioxidant, antimicrobial and antitumoral activities [15].

Considering all the health benefits promoted by the bioactive compounds, the commercial interest of food and pharmaceutical industries on these molecules has been intensified along with the development of new technologies for industrial scale production [15,16]. Soybean meal represents a cheaper and readily available source of bioactive compounds, which could be applied in the isolated form or as a mixture of them using the synergistic effect between bioactive molecules present in soy by-products for different purposes [17].

The main objective of the present study was to quantify health-promoting compounds from soybean oil refinery by-products, providing detailed information about valuable bioactive phytochemicals that can be recovered from extruded soybean material after water extraction, as well as their bioaccessibility. The antioxidant activity, the potential antimicrobial and antitumoral bioactivities in the soluble extract were screened.

## 2. Results

### 2.1. Identification and Quantification of Bioactive Compounds in Soybean Meal Aqueous Extract and Their Bioaccessibility

Isoflavone isoforms 7-*O*-glucosides, 6-*O*-acetyl glucosides and 6-*O*-malonyl glucosides and aglycones, each one composed of 3 compounds totaling 12 isoflavone-derivatives, were found in soybean meal aqueous extract, with a predominance of the glucosidic form of genistin (49 mg/100 g), daidzin (35 mg/100 g) and glycitin (16 mg/100 g), which contributed 100 mg/100 g, compared to malonyl (24 mg/100 g), acetyl (6.6 mg/100 g) and aglycone (8.3 mg/100 g) forms (Table 1).

In addition to the isoflavones, sixteen distinct phenolic compounds were identified in the aqueous extract, six in the 4-hydroxybenzoic acid group, four in the hydroxycinnamic acid group and six within flavones and flavonols, all of them found in the soluble fraction. Naringin, 4-hydroxybenzoic acid, gallic acid, vanillic acid, and siringic acid were found in both soluble and insoluble fractions, where they were extracted by acid or alkaline hydrolysis as already mentioned (Table 2).

The major compounds were vanillic acid (9.1 mg/100 g), hesperidin (9.1 mg/100 g), syringic acid (8.1 mg/100 g), gallic acid (7.7 mg/100 g), caffeic acid (6.1 mg/100 g), hydroxibenzoic acid (5.1 mg/100 g), myricetin (5.0 mg/100 g) and rutin (4.9 mg/100 g).

Two 4-hydroxibenzoic acids, gallic acid and gentistic acid, and four hydrocinnanic acids, 5-caffeoylquinic acid, ferulic acid, *p*-coumaric acid and sinapic acid were described in soybean sprouts, after hydrolysis according to the Phenol-Explorer 3.6 database (Phenol-Explorer, 2018, AFSSA, the University of Alberta, the University of Barcelona, IARC and In Siliflo), and all of them in lower concentrations than quantified in the present study (Table 2).

The content of isoflavones in the aqueous extract (pre-digestion) was 138.9 mg/100 g (dry weight) and 5%, 19% and 29% decreases were observed after salivary, gastric and intestinal digestions, respectively. At colonic fermentation, the amount of isoflavones was 62.9, 47.7 and 29.1 mg/100 g of dry weight at the time intervals of 4, 24 and 48 h of fermentation, respectively (Table 1). However, an increase in the acetyl derivative of isoflavones was observed at the same time that the concentration of the total amount of isoflavones was reduced. The acetyldiadzin was increased 3-fold after 24 h of colonic fermentation while the acetylgenistin and acetylglycitin showed a discreet enhancement (Table 1). The content of glycosides and malonyl isoflavones decreased over the gastrointestinal digestion and these compounds, if present, are in undetectable levels in the colonic digestion phase. 

The total content of phenolic compounds in soybean meal aqueous extract was 73.7 mg/100 g and after intestinal digestion this was reduced to 40.9 mg/100 g (Table 2). A lower release of phenolic compounds was observed after the oral digestion, 27.4%, following an increase in the contents of phenolic compounds after the gastric and intestinal digestions, 49.6% and 55.9%, respectively. Ferulic acid, *p*-coumaric acid and kaempferol showed no changes in their contents at any stage when compared to pre-digested samples. However, hesperidin and 4-hydroxibenzoic acid decreased during the intestinal digestion, while caffeic acid, gallic acid, syringic acid, and vanillic acid increased after duodenal digestion.

After the colonic digestion, none of these phenolic compounds were found in detectable concentration.

### 2.2. Bioactivities of Soybean Meal Aqueous Extract

#### 2.2.1. Antioxidant Activity

Soybean aqueous extract showed a similar ability in inhibiting oxidative degradation, evaluated as inhibition of lipid peroxidation, as seen with natural antioxidants, such as α-tocopherol, or synthetic ones, such as buthylated hydroxyanisole (BHA) and butylated hydroxytoluene (BHT), as showed in Appendix A.

The antioxidant activity of soybean meal aqueous extract was evaluated by five different assays, ferric reducing ability of plasma (FRAP), trolox equivalent antioxidant capacity (TEAC), oxygen radical antioxidant capacity (ORAC) and lipid peroxidation inhibition, and associated with total antioxidant potential (TAP) determinations. The TAP and antioxidant capacity of aqueous extract from soybean meal, determined by FRAP, TEAC, ORAC assays, were 74.46%, 81.23 μmol/ Fe^2+^, 207.67 μmol of Trolox/g and 31.61 μM of Trolox/g, respectively. Through the TAP analysis, it was demonstrated that the extract is able to inhibit 74.46% of the Fe oxidation reaction (Figure 1).

There were no differences in the antioxidant activity in all tests performed after the oral digestion, when compared to pre-digestion. In the gastric fluid, we observed an increase of 69% in the antioxidant activity evaluated by TEAC and 100% in ORAC assays, but TAP was enhanced by 22%, when compared to pre-digestion and oral digestion. In the intestinal digestion, antioxidant ability was maintained when compared to those observed in the gastric fluid, after the ORAC and TAP assays (Figure 1).

#### 2.2.2. Antimicrobial Activity

The aqueous extract of soybean meal was able to abolish the growth of strains from distinct species and genus as observed for *A.* genomospecie 3 ATCC 17922 and *A. hydrophila* FDA 110-36, several *E. coli*s trains - ATCC 43895 and DH5 alfa, *P. fluorescens* ATCC 13525, *S. aureus* ATCC 14458, *coagulase*-negative *Staphylococcus saprophyticus* KT955005, respectively, *V. parahaemolyticus* ATCC 17802 and *Salmonella enterica, strains* ATCC 1225 and ATCC 29934. *L. innocua* ATCC 33090 was less sensitive to the antimicrobial agents from soybean meal aqueous extract since 21% of growth inhibition was observed when exposed to 150 mg/mL of aqueous extract (Table 3).

#### 2.2.3. In Vitro Cytotoxicity Assay in Different Cell Lineages

Different concentrations, from 125 mg/mL to 0.97 mg/mL, of the aqueous extract of soybean meal affected the viability of the three cell lineages in different manners (Figure 2, *top panels*). At concentrations of 0.97 to 3.9 mg/mL, healthy mice bone marrow cells viability was not affected in 24 h. On the other hand, at concentrations above 3.9 mg/mL, BM viability was significantly reduced, exhibiting an IC_50_ of 16.1 mg/mL.

The viability of the mouse fibroblast L929 cell line was not reduced at soybean extract concentrations from 0.97 to 62.5 mg/mL, a stimulation of cell proliferation was seen though, with an IC_50_ of ~72.06 mg/mL. On the other hand, at the highest concentration of 125 mg/mL, L929 cell viability was completely abolished. Similarly, the viability of the human glioblastoma U-87 MG cell line was also completely inhibited at soybean concentration of 125 mg/mL and exhibited and IC_50_ of 65.7 mg/mL. Inferior concentrations of soybean meal aqueous extract did not affect cell viability after 24 h of exposure (Figure 2).

## 3. Discussion

Plant phytochemicals, such as isoflavones and phenolic compounds, can be extracted using different solvents, including ethanol, methanol, acetone, ethyl acetone, or in combination with water in different proportions. Organic solvents are commonly used for the extraction of free and conjugated phenolics, but some cannot be accepted as food grade, due to their toxicity [20,21]. In contrast, when water is used for extraction, the recovered bioactive compounds can be used as food additives without risks for human health.

Isoflavones usually found in the different colored seed coats of soybeans (yellow, black, brown, and green) have been evaluated in crops for two years, and have also been detected in soybean meal aqueous extracts obtained from those soybean seeds [22]. Herein, the crude extract was obtained without the use of organic solvents and exhibited high isoflavone contents, of 138.9 mg/100 g dry weight (Table 1). Although isoflavone contents found in the soybean meal aqueous extract were lower (35%) than the data available for soybean and raw derivatives at the Phenol-Explorer 3.6 database [23], the amount is still meaningful, because the recovery of these phytoestrogens does not include an obligatory step of organic solvent removal.

As a by-product from soybeans, obtained during soybean oil refining, the soybean meal aqueous extract contains the same major isoflavones in the same structural form originally present in soy beans, with high concentrations of isoflavones in the glycosidic form [10], corresponding to 66% of the total (Table 1). Several studies have demonstrated that the thermal treatment carried out during soybean processing promotes the conversion of malonyl glycosides to β-glycosides [24]. Soybean processing into flour promotes the conversion of isoflavones to their aglycones in different ratios, depending on the method used to leach the flour, which facilitates isoflavone absorption [25]. Acetyldaidzin and acetylglycitin found in low concentrations in the aqueous extract were not identified in soy grains, according to the Phenol-Explorer 3.6 database [23] (Table 1). On the other hand, the aqueous extract showed a 3-fold increase of total isoflavones described in soy protein isolate (from 47 to 62 mg/100 g) [9]. Daidizin (35 mg/100 g), glycitin (16 mg/100 g) and genistin (49 mg/100 g) concentrations in the soybean meal aqueous extract described herein were higher than those described by Mujić et al. [26] who evaluated five soy seed cultivars and found daidizin contents ranging from 12.4 to 24.4 mg/100 g, glycitin contents from 5.9 to 11.4 mg/100 g and genistin ranging from 18.6 to 37.1 mg/100 g (Table 1). On the other hand, the amounts of daidzein and genistein found in the soybean meal extract were lower than those reported by Mujić et al. [26], who described 2.5- to 11-fold higher concentrations. It is worth mentioning that among phenolics the flavonoid family, especially genistein, exhibits the highest antioxidant activity, due to its chemical structure, as well as phytoestrogen effects [27].

As expected, no modifications occurred in isoflavone profiles after oral digestion. In the gastric and duodenal phases, a discrete decrease in isoflavone content (17.06% and 9.9%) was observed. However, during intestinal digestion, aglycones content increased, as discussed previously, suggesting that the intestinal environment is adequate for aglycone release from the soybean meal matrix. Glycosylated isoflavones are partially hydrolyzed in the small intestine, releasing aglycones (daidzein, genistein and glycitein), which are then absorbed. Non-hydrolyzed compounds follow to the large intestine where they can be hydrolyzed and, consequently, absorbed [28]. After the colonic phase, acetyl and aglycone forms increased due to the action of enzymes from colonic microflora. However, acetyl glycitine was not detected after 24 h and 48 h of colonic fermentation. Bacterial hydrolysis of isoflavone glycosidic forms can be attributed to β-glycosidases enzymes, which display high affinity for daidzin and genistin, hydrolyzing them to their aglycone forms [29]. 

In addition to isoflavones, 16 other phenolics were identified in the soybean meal aqueous extract, in both soluble and insoluble fractions (after alkaline and acidic hydrolysis). Naringin, 4-hydroxybenzoic acid, gallic acid, vanillic acid and siringic acid were identified (Table 2). Soluble phenolics are predominant in fruits and vegetables and it is important to note that they show higher bioavailability and duodenal absorption in humans, explaining their significant biological importance [30]. The main phenolic compounds identified in soybean and soybean meals were phenolic acids (chlorogenic acid, caffeic acid, ferulic acid, syringic acid, *o*-cummaric acid, *p*-coumaric acid and vanillic acid) and flavonoids [4,31]. In another study, 12 isoflavones and 21 other phenolic compounds were found in defatted soybean flour extracts, including gallic acid, homogentysic acid, protocatecuic acid, chlorogenic acid, catechin, 4-hydroxybenzoic acid, vanillic acid, caffeic acid, syringic acid, *p*-coumaric acid, rutin, ferulic acid, ferric acid, naringin, myricetin, quercetin, *trans*-cinnamic acid, kaempferol, hesperetin, hesperidin and synapic acid [4]. High concentrations of syringic acid, vanillic acid, salicylic acid and benzoic acid were detected in the soybean meal aqueous extract obtained in the present study. However, the fact that phenolic composition and contents in soy grains and soy meals varies according to species, cultivar, and season, geographic and environmental conditions, maturity should be taken into account, as well as the extraction and analysis methods [12,32]. Phenolic aqueous extraction proved to be more efficient, since 60% of the total phenolic compounds were extracted, and most already identified in soybean meals were found in the aqueous extract. Such soybean meal aqueous extracts could be used for several purposes, including food fortification and/or food preservation.

The in vitro bioaccessibility measurements for phenolics observed herein, particularly isoflavones, through the antioxidant activity determinations, support the health benefits of bioactive compounds of nutritional significance [33,34,35]. In one study, the in vitro bioaccessibility estimation of apple phenolics was performed, almost 65% of the compounds were released following gastric digestion, and less than 10% after the intestinal digestions [36]. These data are similar to those observed herein regarding the major release of phenolic compounds after gastric digestion in comparison to intestinal digestion. On the other hand, the percentage of phenolic compounds released from the soybean meal aqueous extract presented a slightly lower percentage, which can be explained by the soybean matrix characteristics, where both free and conjugated phenolics are found, while only free phenolics were estimated in apples. After intestinal digestion, a 32% decrease in hesperidin and 30% decrease in 4-hydroxybenzoic acid contents were observed (Table 2), but these phenolics, upon release, may interact with other constituents of the food matrix, as described for soy beverages [37]. Gallic acid, syringic acid, vanillic acid and myricetin were released during intestinal digestion (25, 29, 29 and 30%, respectively) at higher amounts when compared to gastric digestion. Perhaps most of the phenolic compounds in food matrices are in their conjugated form (ester, glycosides or polymers), which cannot be absorbed directly but become available after hydrolysis by intestinal enzymes on the brush border [38].

Most phenolic compounds present in the soybean meal aqueous extracts are in their conjugated form and require hydrolysis before being absorbed [38]. Data obtained after oral digestion indicates that approximately 20% of the compounds in the soybean meal aqueous extract are found in their free form, since no modifications and compound release are expected. During gastric and intestinal digestions, a considerable release of phenolic compounds was observed, except for ferulic acid, *p*-coumaric acid and kaempferol, which did not present alterations when compared to their pre-digestion content. In some cases, phenolic compounds, such as caffeic acid, gallic acid, vanillic acid and rutin, increased after the gastric and intestinal digestion phases. Other phenolic compounds, such as 4-hydroxybenzoic acid, 5-caffeoylquinic chlorogenic acid and hesperidin, increased significantly after the gastric phase but decreased after the intestinal phase. These results are in agreement with some studies that concluded that the stability and antioxidant activity of phenolic compounds depend on the physicochemical conditions of the digestive tract segments, such as pH, temperature and enzyme activities, as well as the nature of the food matrix [36]. Racemization of phenolic compounds occurs, altering and rendering more reactive compounds compared to the beginning of the digestive process. On the other hand, the alkaline pH during intestinal digestion may reduce racemization (forming fewer reactive compounds) and degrade phenolic compounds, leading to loss of activity and bioaccessibility [39]. Furthermore, some phenolic compounds are sensitive to the alkaline pH in duodenal fluid, and a portion may be modified into distinct structures, probably to their non-detectable aglycone forms, facilitating absorption through epithelial cells from the duodenal mucosa [38]. No extraction step of these bioactive compounds from the soybean meal aqueous after each phase of the in vitro human simulated gastrointestinal digestion was performed. After each stage of the in vitro simulated digestive process (oral, gastric or intestinal), an aliquot was removed (from each stage) for isoflavone and phenolic compound analyses. The aim was to assess bioaccessibility, i.e., the fraction or content of phenolic compounds released from the food matrix (soybean meal aqueous extract), in the gastrointestinal tract during in vitro gastrointestinal digestion, simulating the bioacessibility of these compounds in the human gastrointestinal tract.

After the digestibility tests—oral, gastric and intestinal digestion—although structural changes might occur in antioxidant compounds and the bioaccessibility of the individual antioxidant compounds from the aqueous soybean meal extract, no difference in the antioxidant overall activity, evaluated by the distinct assays—TAP, ORAC, FRAP and TEAC- was observed, when compared to the pre-digestion state (Figure 1). In a previous study, antioxidant capacity determinations of soybean meal and soybean grain were carried out, showing an average amount of 7.2 µmol Fe^2+^/g and 11.5 µmol Trolox/g, respectively, for soybean grain and 10.3 µmol Fe^2+^/g and 18.6 µmol Trolox/g for soybean meal, considering the FRAP and TEAC assays, respectively [40]. However, after aqueous extraction of the same sample, approximately 8-fold and 11-fold increases in FRAP and TEAC antioxidant activities were found, of 81.23 µmol Fe^2+^ and 207.67 µmol Trolox/g, respectively. Xu & Chang [41] evaluated the antioxidant capacity of 30 different soybean cultivars and found an average activity ranging from 21.2 to 119.3 μmol ORAC, whereas the activity described herein for the agro-industrial by-product was of 31.61 μmol Trolox/g. This higher antioxidant activity may be due to the aqueous extraction process, that includes a post-heat treatment, which may have released bioactive molecules from soybean meal matrix. The use of the soybean meal aqueous extract as a source of phenolics seems to be promising due to the presence of several molecules that can act in synergy with each other, promoting and enhancing the antioxidant capability of the material, while also presenting food grade. 

Moreover, similar to other phenolics, soybean isoflavones can either be modified during gastrointestinal tract digestion as mentioned previously, involving thermic treatment and microbial fermentation, contributing to the increased antioxidant capacity of the compounds [42] or, as seen for phenolics, might be conjugated, and cannot be absorbed directly but only after becoming available [38,42]. An increase in antioxidant activity was observed in the gastric fluid after all assays, since acid hydrolysis seems to be capable of releasing phenolic compounds linked to lignins by ether bonds through their hydroxyl groups on the aromatic ring and ester bonds with proteins and structural carbohydrates via their carboxylic group (Figure 1). The acid treatment mainly breaks down glycosidic bonds and solubilizes sugars [30], thus releasing phenolic compounds and increasing antioxidant activity. In addition to total phenolic compounds, isoflavone content, profile and activity varied during the gastrointestinal digestion processes, which included physico-chemical processes such as hydration, mild heat treatment, and microbial fermentation, increasing the antioxidant power of these molecules in general [42]. 

The aqueous soybean meal extract can substitute natural antioxidants, such as α-tocopherol, or synthetic ones, such as BHA and BHT, in inhibiting oxidative degradation against lipid peroxidation, as described in Appendix A. The aqueous soybean meal extract showed similar antioxidant activities to synthetic antioxidants, BHT and BHA, commonly used as preservatives by the food industry and comparable to α-tocopherol, known as the most efficient natural chain-breaking free radical reaction. Natural antioxidants can play a key role in food preservation, delaying oxidation-induced deterioration, and they may substitute many synthetic food preservatives, since their use is strictly regulated due to carcinogenic effects in animals [43]. Even in physiological conditions, such as the digestive process, gastrointestinal tract tissues are constantly exposed to newly formed reactive oxygen species, and antioxidant compounds may play a role in maintaining the redox balance against harmful oxidants and preventing gastrointestinal tract diseases linked to ROS generation [38,43].

Despite the methods used for the reduction or elimination of foodborne pathogens, an increase in foodborne diseases, a major threat to public health, is still noted. Due to consumer awareness, natural antimicrobials have been replacing antibiotics in food and their derivatives and the search for natural antimicrobials, mainly derived from plants, has increased [44]. The soybean meal aqueous extract was able to inhibit the growth of both Gram-positive and Gram-negative bacteria (Table 3). The antimicrobial activity of the soybean meal aqueous extract can be due, at least partially, to the oligopeptide-rich fraction, with molecular masses lower than 14 kDa, as described previously. Additionally, phenolic compounds detected in the aqueous extract can also generate adverse microbial growth effects, by inhibiting microbial metabolic functions [44,45]. The effects observed herein could be a result of the synergistic action of both groups of molecules. 

Studies on the growth inhibition of foodborne pathogens tested against phenolic extracts prepared from soybean flour indicate that Gram-positive bacteria are more sensitive than Gram-negative bacteria [4], but in the present study both bacteria displayed sensitivity to the aqueous extract. Gram-negative bacteria are more resistant to antimicrobials due to their outer membrane, composed of lipopolysaccharides, which are not found in Gram-positive bacteria. However, in the present study, the seven tested Gram-negative bacteria strains were inhibited, indicating that differences in phenolic sensitivity is due to outer membrane proteins that might aid in phenolic permeability from the soybean meal aqueous extract [44]. These results have prompted the food industry to search for alternative food preservatives that can enhance food safety and quality. Compounds derived from natural sources display the potential to be used for food preservation, due to their antimicrobial properties against a broad range of foodborne pathogens. 

The evaluation of putative cell injury or cytotoxicity from phytochemicals and/or toxins found in soybean meal aqueous extract were assayed against healthy bone marrow, fibroblasts and tumoral cell lines. Extract characterization and the evaluation of the cytotoxic effects of compounds of plant origin and the understanding of the potential benefits and/or toxicity of these plants to health are of fundamental importance to reduce the possible risks of these agents. 

The cytotoxicity of the aqueous soybean meal depends on the applied concentration and cell lineage. In the concentration range from 0.97 to 3.9 mg/mL the soybean meal aqueous extract did not affect the viability of healthy mouse bone marrow cells. On the other hand, cellular toxicity was observed when higher concentrations were used, indicating that the use of this aqueous soybean meal extract in food products should be previously assessed in healthy human cell lines, since higher concentrations could lead to cytotoxic effects to these healthy cells, as observed for murine cells. For this purpose, bioaccessibility as well as bioavailability should be considered, in order to determine the maximum concentration of the soybean meal aqueous extract that could be used without causing toxicity to healthy cells. It is widely known that plant extracts are composed of different varieties of compounds other than polyphenols, which could contribute to cytotoxicity against healthy cells. Further studies should be performed to determine the cytotoxicity of each isolated compound identified in the soybean meal aqueous extract. 

Similarly to the bone marrow cells, another healthy murine cell line, L929 fibroblasts, was not affected at concentrations ≤62.5 mg/mL, exhibiting an IC_50_ of ~72.06 mg/mL, but its growth was completely inhibited at 125 mg/mL, reinforcing the non-cytotoxic effect of soybean meal aqueous extract to healthy cells in a dose-dependent manner. The U-87 MG glioblastoma cell line viability was completely eliminated at soybean meal aqueous extract concentrations of 125 mg/mL. As the effective concentration does not match concentrations considered safe for healthy bone marrow cells, cytotoxicity tests should be carefully evaluated, as most antitumoral compounds in clinical use exhibit some level of toxicity against healthy cells. However, the administration regimen can overcome undesirable effects. Further investigations should be conducted to determine whether the antitumoral effect on U-87 MG cells at 125 mg/mL resulted from specific phenolic compounds and/or peptide from soybean meal aqueous extract interactions with tumor cell receptors.

It is widely known that phenolic acids and flavonoids, such as caffeic acid derived from hydroxycinnamic acids found in the soybean meal aqueous extract display antitumoral activity [30]. Cell viability decreases could be caused by the structural and functional damages in the assessed cell lineage [46]. Additional studies are required to determine what types of and the extension of the damage that the soybean meal aqueous extract caused to tumoral and healthy mouse BM and L929 cells.

## 4. Materials and Methods

### 4.1. Organisms

Samples of soybean meal (*Glycine max*) (9.6 g/100 g moisture) were donated by a soybean oil crush Brazilian industry. Antimicrobial analysis were performed by using the following microorganisms: *Acinetobacter* genomospecies 3 (isolated from a sludge refinery), *Pseudomonas fluorescens* ATCC 13525, *coagulase*-negative *Staphylococcus saprophyticus* KT955005 (isolated from soft cheese), *Listeria innocua* ATCC 33090, *Staphylococcus aureus* ATCC 14458, *Aeromonas hydrophila* strains FDA110-36 and ATCC 7966, *Escherichia coli* strains ATCC 43895 and DH5 α, *Salmonella enterica* subspecies *diarizonae* strains ATCC 12325 and ATCC 29934 and *Vibrio parahaemolyticus* ATCC 17802 were kindly provided by the Oswaldo Cruz Institute - INCQS cell bank.

Cytotoxicity evaluation was estimated by cell viability assays using three cell lines, mouse bone marrow and fibroblast L929 cells (Sigma-Aldrich Co, St. Louis, MO, USA) and human glioblastoma U-87 MG cell line (Sigma-Aldrich Co, St. Louis, MO, USA).

### 4.2. Preparation of Soybean Meal Aqueous Extract

The aqueous extract was prepared according to Del Aguila et al. [45] where 50 g of triturated soybean meal (extruded soybean material) was homogenized in 150 mL of distilled water followed by incubation at 50 °C for 24 h in constant agitation to activate endogenous protease activity. The resulting suspension was centrifuged at 8000× *g* for 10 min at room temperature, and the supernatant was treated at 90 °C for 10 min with constant agitation, to stop protease activity, followed by filtration in a 0.22 µm pore membrane (Merck Millipore Co, Feldbergstraße, Darmstadt, DE).

### 4.3. Identification and Quantification of Isoflavones Forms by LC-DAD-FLD

Isoflavones forms were analyzed by LC-DAD-FLD as described by Fonseca et al. [47] by using a quaternary pump (LC-10AD *vp*) chromatographic system LC-20A Prominence (Shimadzu, Kioto, JP) with a column oven (CTO-10AS *vp*), a handgun (8125 Rheodyne, equipped with a loop in 100 µL) a diode arrangement detector (DAD), fluorescence detector (FLD) and a C18-5 μm Kromasil^®^ (250 × 4.6 mm) column maintained at 40 °C. The mobile phase was milli-Q water (eluent A) and acetonitrile 100% (eluent B), both added with 0.3% formic acid at a flow rate of 1.0 mL/min. Before injection, the column was equilibrated with a mixture of eluent A and B (85%:15%) and then the eluent mixture was modified during the 30 min assay. Injection intervals of 25 min were used to re-equilibrate the column with the initial gradient. Quantification was performed by external standardization using a diode array detector (DAD) set to 190–400 nm and fluorescence excitement at 280 nm and emission at 310 nm.

Quantification was performed by external standardization. Isoflavones were quantified by DAD peak area at 250 nm. The contents of malonyl glycosides and acetyl glycosides isoflavones were determined from the calibration curve of the corresponding β-glycoside isoflavone, correcting for differences in molecular weight. The results were expressed as mg of compound per 100 g of dry weight of extract (dwb). Isoflavone quantification was performed using a calibration curve (0.1–10.0 ppm) with standards at a minimum of five levels of concentration (Appendix A).

### 4.4. Identification and Quantification of Phenolic Compounds (PC)

PC extraction from the soybean meal extract was performed in triplicate, according to Inada et al. [48] using two extraction methods. Soluble PCs were extracted by ethanol: H_2_O-DD (80:20, *v*/*v*) for 10 min and centrifuged at 2500× *g* at 10 °C for 5 min. The extraction was performed twice, the supernatants were combined, the solvents were removed in a rotary evaporator at 130 rpm and the dried residue was reconstituted in H_2_O-DD.

The PC conjugate extraction was carried out by alkaline and acid hydrolysis, where alkaline hydrolysis was performed in 10 M NaOH and H_2_O-DD added to residues after soluble PCs extraction. The residues were kept under stirring for 16 h at room temperature, preventing light exposure. Subsequently, the residue was adjusted to pH 2 with concentrated HCl and then extracted with ethyl acetate. After centrifugation at 2500× *g* for 5 min at 10 °C, the supernatants were treated by alkali, repeated twice. Supernatants were combined, solvents were removed and the dried residues were reconstituted in methanol: H_2_O-DD (80:20 *v*/*v*). 

The acid hydrolysis was performed by adding 100% HCl to the residues following alkaline extraction. The residues were kept in a water bath at 85 °C for 30 min, and the extraction with ethyl acetate was performed. 

All extracts were filtered through 0.45 µm pore cellulose ester-membranes (Merck Millipore Co, Feldbergstraße, Darmstadt, DE) prior to the HPLC analyses.

The HPLC device was equipped with a 5 μm reverse phase C18 column (250 × 4.6 mm, I.D., Ascentis^®^, LA, USA) guarded by a 5 μm C18 guard column (10 × 3.0 mm, I.D., Ascentis^®^) and with a diode-array detector (DAD) detector model SPD-M30A (Shimadzu, Kioto, JPN). The DAD wavelength was monitored from 190 nm to 370 nm. Column temperature was set at 40 °C and the injection volume was 20 µL for all samples. The mobile phase (1.0 mL·min^−1^) was 0.3% formic acid (in H_2_O-DD), methanol (100%) and acetonitrile (100%) at a gradient elution. The column was equilibrated with 81% formic acid, 18% methanol and 1% acetonitrile. After sample injection, formic acid and methanol concentrations were raised to 79% and 20%, respectively, in 1 min; 56% and 43% in 18 min; 14% and 85% in 23 min, and kept constant up to 30 min. Between injections, the column was re-equilibrated with 81% formic acid, 18% methanol and 1% acetonitrile for 10 min. PCs quantification were performed using a calibration curve (1–50 ppm) with standards at a minimum of five levels of concentration (Appendix A).

### 4.5. Evaluation of the Antimicrobial Activity

Bacteria, obtained from an inoculum suspension at 10^6^ cells/mL, were grown in the appropriate medium, following the recommended instructions provided by the guide manual from the cell bank. *A.* genomospecie 3 ATCC 17922, *E. coli* DH5 alfa, ATCC 43895 and *P. fluorescens* ATCC 13525 grown in LB media (Luria-Bertani) (BD^™^, Le Pont de Claix, FRA), *S. aureus* ATCC 14458, coagulase-negative *S. saprophyticus* KT955005, and *A. hydrophila* FDA110-36 grown in BHI (Brain Heart Infusion) (BD^™^), *V. parahaemolyticus* ATCC 17802 and *Salmonella enterica* ATCC 1225 and ATCC 29934 were grown in NB (nutrient broth) (Himedia, Mumbai, IND). Antimicrobial activity was evaluated by the broth macro-dilution method [49] with modifications. Bacteria were grown in the absence (negative control) and presence of increasing concentrations of the soybean meal aqueous extract, from 12.5 to 150 mg/mL, and incubated at 37 °C for 18 h. After incubation, the bacteria suspension was serial diluted 1/10 in saline 0.85 % of NaCl, 0.2% Tween 80) and spread on LB, NB or BHI solid media. Plates were incubated at 37 °C for 18 h and colony-forming units (CFU) counted. The procedures were performed in triplicate and the lowest concentration able to prevent visible bacteria growth was recorded as the minimal inhibitory concentration (MIC).

### 4.6. Determination of Total Antioxidant Activity

#### 4.6.1. Determination of FRAP

The FRAP assay was carried out by a modification of the method described by Benzie & Strain [50]. Soybean meal extract samples were diluted (1:10) and then mixed thoroughly with the FRAP reagent. Standard FeSO_4_ solutions were used and absorbances at 593 nm were measured in a V–530 UV/VIS spectrophotometer (Jasco^®^, MD, USA). FRAP results for each sample were evaluated in triplicate and expressed as mmol of Fe^II^/100 g.

#### 4.6.2. Determination of TEAC

TEAC assays were performed using a modification of the method described by Re et al. [51]. The ABTS radical cation (ABTS^•+^) was generated by a chemical reaction of ABTS with K_2_S_2_O_8_ in darkness at room temperature for 12–16 h. For the sample analyses, soybean meal extract sample was mixed with the ABTS^•+^ reagent and absorbances at 720 nm were measured in a V-530 UV/VIS spectrophotometer (Jasco^®^, MD, USA). TEAC were determined in triplicate and related to the ABTS^•+^ inhibition percentage by antioxidants in the samples. TEAC results for soybean meal extract sample were evaluated in triplicate and expressed as mmol of Trolox/100 g.

#### 4.6.3. Determination of TAP

Soybean meal aqueous extract was analyzed as previously described by da Silva et al. [52]. Samples were diluted (1:10) and centrifuged at 4500× *g* for 10 min and the supernatants were filtered through a 0.45 µm pore cellulose membrane (Merck Millipore Co, Darmstadt, GER). The resulting samples were transferred to amber vials and incubated at 37 °C for 10 min with a solution containing 1 mM Fe^2+^, 10 mM H_2_O_2_ and 1 mM terephthalic acid (TPA) in 50 mM phosphate buffer (pH 7.4). The hydroxyterephthalic acids (HTPA) were detected by HPLC. TAP measurements were obtained by the difference between the chromatogram surface area generated in the Fenton reaction with and without the sample (Appendix A).

#### 4.6.4. Determination of ORAC

The ORAC assay was performed according to Zulueta et al. [53] with modifications. Sample absorbances were determined on a Wallac 1420 VICTOR multilabel counter (Perkin–Elmer Inc., Hopkinton, MA, USA) with fluorescence filters at an excitation wavelength of 485 nm and emission wavelength of 535 nm. A stock solution of fluorescein was prepared by weighing 3 mg of fluorescein followed by dissolution in in 100 mL of phosphate buffer (75 mM, pH 7.4). The stock fluorescein solution was stored in complete darkness under refrigeration conditions. Then, the fluorescein working solution (78 nM) was prepared daily by dilution of 0.100 mL of the fluorescein stock solution in 100 mL of phosphate buffer. The AAPH radical (221 mM) was prepared daily by mixing 600 mg of AAPH in 10 mL phosphate buffer. A 25 µM Trolox solution was used as reference standard, prepared daily in phosphate buffer from a 4 mM stock standard solution kept in the freezer at 20 °C. A total of 100 μL of fluorescein (78 nM) and 100 μL of the samples, blanks (phosphate buffer), or standards (25 µM of Trolox) were added to each well, followed by 50 μL of AAPH (221 mM). ORAC values, expressed as μM Trolox equivalents were calculated by applying the following formula:
ORAC (μM Trolox equivalents) = C_Trolox_·(AUC_Sample_ − AUC_Blank_) *k*/(AUC_Trolox_ − AUC_Blank_)(1)
where C_Trolox_ is the concentration (µM) of Trolox, *k* is the sample dilution factor, and AUC is the area below the fluorescence decay curve of the samples, blanks and Trolox, respectively, calculated using the GraphPad Prism v.5 software package (GraphPad Software Inc., San Diego, CA, USA). ORAC determinations were performed in triplicate and values were expressed as mmol Trolox equivalents/100 g.

### 4.7. Lipid Peroxidation Inhibition

Lipid peroxidation analyses were carried out as described by Samaranayaka et al. [54]. BHA, BHT, α-tocoferol and soybean meal aqueous extract samples at a final assay concentration of 0.2 mg/mL were prepared in an emulsion system containing linoleic acid (1.3% in ethanol)/ethanol (75% /H_2_O-DD). Flasks containing samples and standards were sealed and incubated at 40 °C at 150 rpm for 7 days without light interference. Aliquots of each flask were taken, in triplicate, at different time intervals—0, 18, 42, 90 and 162 h. A solution containing ethanol (75%), ammonium thiocyanate (30%) and 20 mM ferrous chloride in 3.5% HCl was added to each flask, and after 3 min, absorbance was measured at 550 nm in DU-530 spectrophotometer (Beckman Coulter Inc., Brea, CA, USA). 

### 4.8. In Vitro Simulated Gastrointestinal Digestion

Analyzes of phenolic compounds, isoflavones, TAP and antioxidant activity assays were repeated following in vitro simulated gastrointestinal digestion performed according to Oomen et al. [55] and Sagratini et al. [56]. Fluids generated after OD, GD and ID were collected and filtered. Phenolic compounds in the permeated suspensions were analyzed by HPLC-DAD and compared to PD.

The ex vivo colonic fermentation was performed according to the methodologies described by Hu et al. [57] and Mosele et al. [58] with modifications. The feces of the volunteers were homogenized in a nutrient-rich medium (0.5 g in 10 mL) described by McDonald et al. [59] and 5.0 mL of this mixture was added to the in vitro digested material at the end of the entire simulated digestion process, incubated at 37 °C and 50 rpm for 48 h. All steps were performed in triplicate.

Feces were donated by volunteers who filled the inclusion criteria: age between 18 and 35 years, body mass index within eutrophy (18.5 to 24.9 kg/m^2^), no gastrointestinal disease, regular bowel function and without the use of nutritional supplements, antibiotics, probiotics, prebiotics or symbiotic in the three months prior to collection of feces. The volunteers were instructed not to avoid for at least two days prior to feces collection, the ingestion of foods rich in phenolic compounds as fruits in general, black beans, cabbage, juices, yogurts, soft drinks, alcoholic beverages, soy and soy-derived products. 

The study protocol was approved by the Ethics and Research Committee (under N°512847) of the University Hospital Clementino Fraga Filho/Universidade Federal do Rio de Janeiro.

### 4.9. In Vitro Cytotoxicity Assays Against Murine and Human Cells

The cytotoxicity of soybean meal aqueous extract was performed by in vitro assays against healthy and tumoral cells. Bone marrow cells were collected from BALB/c mice aged 8–10 weeks. The study protocol was approved by the Institutional Ethics Committee for Animal Research/ Universidade Federal Fluminense under N° 821-16. 

One hundred microliters containing healthy mouse bone marrow (BM) cells (5.0 × 10^5^ cells/mL) or fibroblast (1.5 × 10^5^ cells/mL) L929 or human glioblastoma U-87 MG cells were cultured in 96-well microplates for 24 h at 37 °C in 5% CO_2_ humidified atmosphere. To a semi-confluent cell layer, 100 μL of soybean meal aqueous extract in concentrations ranging from 125 to 0.97 mg/mL were added to the wells and after 24 h, cell viability was evaluated by the addition of 20 µL of resazurin (2.5 µg/mL) (Sigma-Aldrich Co), according to McMillian et al. [60] with modifications. Cells were incubated in the presence of resazurin for an additional 6 h following fluorescence intensity measurement in a Victor™ X microplate reader (Perkin Elmer Inc., Waltham, MA, USA) at excitation and emission wavelengths of 530 and 590 nm, respectively.

### 4.10. Statistical Analysis

Two-way analysis of variance (ANOVA) with repeated measurements was performed to identify differences in isoflavones, phenolic compounds and antioxidant capacity among soybean meal aqueous extract before and after the in vitro simulated digestion. Statistical significance was set at the confidence level of 0.05 and additional post hoc test with Bonferroni adjustments were performed. Values were expressed as mean ± standard deviation (SD) and statistical procedures were completed using the GraphPad Prism v.5 software package (GraphPad Software Inc., San Diego, CA, USA). 

Cell viability assays were carried out in triplicate and evaluated by a one-way ANOVA with Tukey post-test to perform all multiple comparisons analyses. Differences were considered to be significant when *p* < 0.05, as determined by using the GraphPad v.7 Software (GraphPad Software Inc., San Diego, CA, USA).

## 5. Conclusions

Soy presents a strategic potential in food safety, as well as being a source of proteins and bioactive compounds for human needs. Isoflavones and phenolic compounds were recovered after aqueous extraction of soybean meal, a byproduct of soybean oil refining. The predominant isoflavones were genistin, daidzin, glycynin and malonylgenistin and their glycosidic forms exhibited an overall bioaccessibility approaching 75%. Sixteen phenolic compounds with high bioaccessibility were identified, with gallic acid, syringic acid, vanilic acid, hydroxibenzoic acid, myricetin, hesperidin and rutin as the most predominant.

The aqueous soybean meal extract was able to inhibit lipid peroxidation similarly to two synthetic antioxidants and a natural antioxidant, α-tocopherol, commonly used by the food industry. The antioxidant capacity evaluation of the aqueous soybean meal extract by different methods indicated that the antioxidant compounds present in this agroindustrial by-product maintained their functionality, even after industrial processing for soybean oil production, and were able to maintain oxidation process inhibition.

The aqueous extract did not present cytotoxicity to healthy murine cells, but presented bioactivities such as antioxidant capacity, which increased after in vitro gastrointestinal digestion, lipid peroxidation inhibition when compared to natural and synthetic antioxidants, and antimicrobial activity against both Gram-positive and Gram-negative foodborne pathogens. The results described herein point to the potential usefulness and novel applications for defatted soybean waste. Compounds derived from the aqueous soybean meal extract have the potential to be used as health promoting agents to fortify or enrich food matrices or to be included as natural additives for the control of foodborne pathogen growth or even as adjuvants to control tumoral cell proliferation.

## Figures and Tables

**Figure 1 molecules-24-00074-f001:**
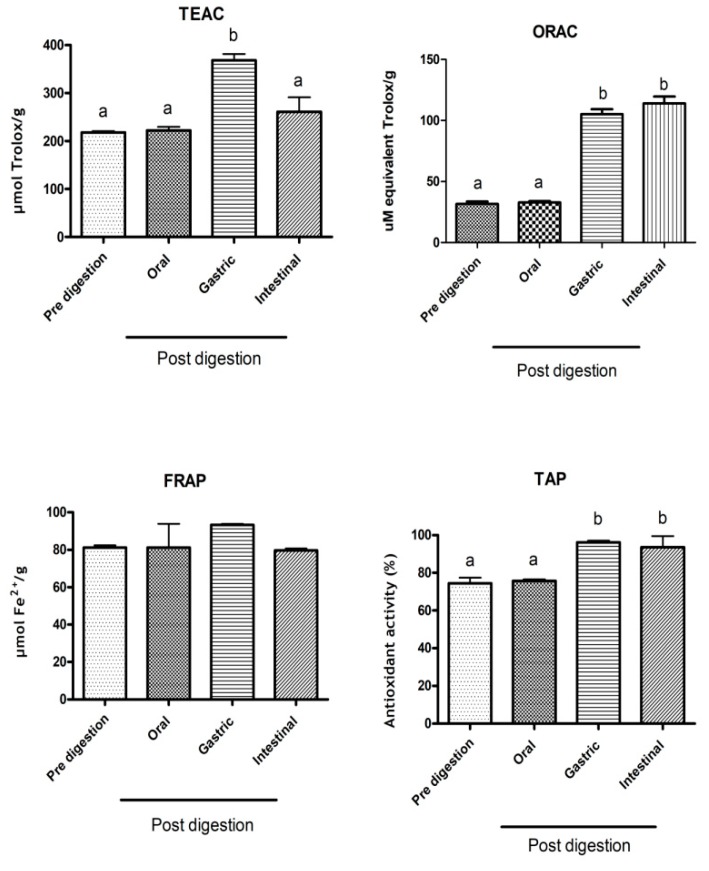
Antioxidant activity following in vitro simulated gastrointestinal digestion. Antioxidant capacity (TEAC), ferric reducing ability in plasma (FRAP), total antioxidant potential (TAP) and oxygen radical antioxidant capacity (ORAC) from aqueous soybean meal extract, after in vitro simulated gastrointestinal (oral, gastric and intestinal) digestion. Values are expressed as the mean ± standard deviation (*n* = 3). Different letters denote difference between means of each antioxidant activity determinations before and after in vitro gastrointestinal digestion (one-way ANOVA, Bonferroni’s post-test), where *p* < 0.05. FRAP assays did not exhibit differences.

**Figure 2 molecules-24-00074-f002:**
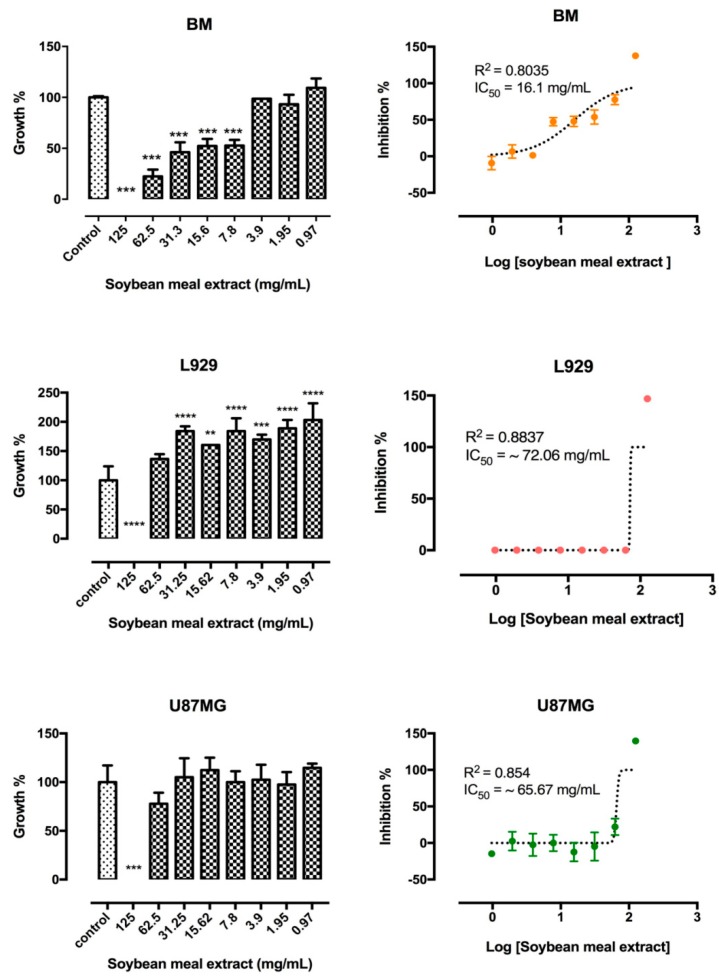
Cytotoxicity of soybean meal aqueous extract on mouse bone marrow (BM) cells and fibroblast L929 cell lines and on the human glioblastoma U-87 MG cell line. Growth versus soybean meal aqueous extract concentration plot (left-hand panel) after 24 h exposure. Right-hand panel shows the inhibition versus Log of soybean meal aqueous extract concentration after 24 h exposure. Results were expressed as the mean ± standard deviation (*n* = 3). Differences between the untreated cells (control) and those incubated with concentrations from 125 mg/mL to 0.97 mg/mL of soybean meal aqueous extract were evaluated by the One-way ANOVA test, with the Tukey post-test where ** *p* < 0.01; *** *p* < 0.001; **** *p* < 0.0001.

**Table 1 molecules-24-00074-t001:** Bioaccessibility of isoflavones from soybean meal aqueous extract estimated by in vitro simulated gastrointestinal digestion (mg/100 g).

Gastrointestinal Digestion (mg/100 g of Dry Weight)					Colonic Fermentation (mg/100 g of Dry Weight)		
**Isoflavones structures**	**PD**	**OD**	**GD**	**ID**	**4 h**	**24 h**	**48 h**
**6-*O*-Acetyl**							
Acetyldaidzin	3.0 ± 0.02 ^c^	3.0 ± 0.02 ^c^	1.3 ± 0.01 ^d^	1.2 ± 0.01 ^d^	1.8 ± 0.02 ^d^	10.1 ± 0.02 ^a^	6.8 ± 0.02 ^b^
Acetylgenistin	3.0 ± 0.01 ^a^	2.0 ± 0.02 ^b^	2.0 ± 0.01 ^b^	2.5 ± 0.03 ^b^	2.7 ± 0.01 ^a^	3.4 ± 0.01 ^a^	3.5 ± 0.02 ^a^
Acetylglycitin	0.6 ± 0.01 ^a^	0.3 ± 0.02 ^a^	0.3 ± 0.01 ^a^	0.2 ± 0.03 ^a^	0.8 ± 0.01 ^a^	nd	nd
**Total**	**6.6**	**5.3**	**3.6**	**3.9**	**5.3**	**13.5**	**10.3**
**Aglycones**							
Daidzein	3.0 ± 0.01 ^d^	2.0 ± 0.01 ^e^	1.1 ± 0.01 ^f^	4.0 ± 0.01 ^d^	21.1 ± 0.05 ^a^	12.7 ± 0.04 ^b^	7.1 ± 0.01 ^c^
Genistein	5.0 ± 0.03 ^b^	3.0 ± 0.01 ^c^	2.0 ± 0.02 ^d^	2.2 ± 0.02 ^d^	11.8 ± 0.05 ^a^	5.8 ± 0.02 ^b^	4.6 ± 0.01 ^b^
Glycitein	0.3 ± 0.01 ^d^	0.2 ± 0.01 ^d^	1.0 ± 0.01 ^e^	1.9 ± 0.01 ^e^	24.7 ± 0.06 ^a^	15.7 ± 0.05 ^b^	7.1 ± 0.01 ^c^
**Total**	**8.3**	**5.2**	**4.1**	**8.1**	**57.6**	**34.2**	**18.8**
**7-*O*-Glucoside**							
Daidzin	35.0 ± 0.02 ^a^	34.0 ± 0.01 ^a^	33.0 ± 0.02 ^a^	10.0 ± 0.01 ^b^	nd	nd	nd
Genistin	49.0 ± 0.02 ^a^	48.0 ± 0.02 ^a^	48.5 ± 0.01 ^a^	11.0 ± 0.01 ^b^	nd	nd	nd
Glycitin	16.0 ± 0.03 ^a^	15.0 ± 0.01 ^a^	14.5 ± 0.02 ^b^	4.0 ± 0.03 ^c^	nd	nd	nd
**Total**	**100.0**	**97.0**	**96.0**	**25.0**	**-**	**-**	**-**
***6-O-Malonyl***							
Malonyldaidzin	7.0 ± 0.03 ^a^	6.0 ± 0.03 ^a^	3.0 ± 0.001 ^b^	1.0 ± 0.03 ^c^	nd	nd	nd
Malonylgenistin	13.0 ± 0.01 ^b^	16.0 ± 0.02 ^a^	5.0 ± 0.001 ^c^	2.0 ± 0.04 ^d^	nd	nd	nd
Malonylglycitin	4.0 ± 0.01 ^a^	3.0 ± 0.03 ^a^	1.0 ± 0.001 ^b^	0.3 ± 0.03 ^c^	nd	nd	nd
**Total**	**24.0**	**25.0**	**9.0**	**3.3**	**-**	**-**	**-**
**Total Isoflavones**	**138.9**	**132.5**	**112.7**	**40.3**	**62.9**	**47.7**	**29.1**

PD, Pre-digestion; OD, oral digestion; GD, gastric digestion and ID, intestinal digestion. Values are expressed as the mean ± standard deviation (*n* = 3). ^a, b, c, d, e^ Different letters within the same line indicate differences between in vitro digestion and colonic fermentation of a compound at significance level *p* < 0.05. nd, not detected. Soybean meal aqueous extract was submitted to in vitro gastrointestinal digestion, simulating the oral, gastric and intestinal phases based on human physiology.

**Table 2 molecules-24-00074-t002:** Bioaccessibility of phenolics from soybean meal aqueous extract.

Phenolic Compounds	PD	OD	GD	ID
	mg/100 g of Dry Weight			
**Hydroxybenzoics**				
Salicylic acid	3.8 ± 0.01 ^a^	2.0 ± 0.2 ^b^	2.0. ± 0.01 ^b^	2.1 ± 0.2 ^b^
Caffeic acid	6.1 ± 0.1 ^a^	2.3 ± 0.2 ^d^	3.0 ± 0.2 ^c^	3.8 ± 0.3 ^b^
Ferulic acid	0.3 ± 0.02 ^b^	0.9 ± 0.01 ^a^	0.9 ± 0.01 ^a^	1.0 ± 0.02 ^a^
Gallic acid	7.7 ± 0.1 ^a^	1.1 ± 0.0 ^d^	2.9 ± 1.3 ^c^	4.8 ± 0.30 ^b^
Syringic acid	8.1 ± 0.1 ^a^	1.1 ± 0.01 ^d^	2.0 ± 0.3 ^c^	4.3 ± 0.3 ^b^
Vanillic acid	9.1 ± 0.2 ^a^	1.3 ± 0.02 ^d^	3.6 ± 0.6 ^c^	6.2 ± 1.6 ^b^
4-hydroxybenzoic acid	5.1 ± 0.1 ^a^	1.2 ± 0.1 ^c^	3.1 ± 0.9 ^b^	1.7 ± 0.4 ^c^
**Total**	**40.2**	**9.9**	**17.5**	**23.9**
**Hydroxycinnamics**				
5-caffeoylquinic chlorogenic acid	3.5 ± 0.2 ^a^	1.0 ± 0.1 ^c^	2.4 ± 0.3 ^b^	1.3 ± 0.1 ^c^
*p*-coumaric acid	2.0 ± 0.1 ^a^	1.0 ± 0.01 ^b^	1.1 ± 0.1 ^b^	1.2 ± 0.1 ^b^
4-hydroxyphenylacetic acid	3.4 ± 0.1 ^a^	1.0 ± 0.1 ^c^	1.6 ± 0.1 ^b^	1.0 ± 0.1 ^c^
Sinapic acid	2.7 ± 0.1 ^a^	1.0 ± 0.1 ^b^	1.1 ± 0.1 ^b^	1.0 ± 0.1 ^b^
**Total**	**11.6**	**4.0**	**6.2**	**4.5**
**Flavonones and Flavonols**				
Hesperidin	9.1 ± 0.4 ^a^	1.2 ± 0.2 ^d^	7.1 ± 0.6 ^b^	4.2 ± 0.6 ^c^
Kaempferol	0.4 ± 0.01 ^a^	0.9 ± 0.1 ^a^	0.9 ± 0.1 ^a^	0.9 ± 0.1 ^a^
Myricetin	5.0 ± 0.01 ^a^	1.3 ± 0.1 ^c^	1.7 ± 0.1 ^c^	3.2 ± 0.6 ^b^
Naringin	2.5 ± 0.5 ^a^	0.9 ± 0.1 ^c^	1.0 ± 0.1 ^c^	1.6 ± 0.8 ^b^
Rutin	4.9 ± 0.1 ^a^	1.8 ± 0.1 ^c^	2.1 ± 0.9 ^b^	2.6 ± 0.1 ^b^
**Total**	**21.9**	**6.1**	**12.8**	**12.5**
**Total phenolics**	**73.7**	**20.0**	**36.5**	**40.9**

Values are the sum of soluble phenolics plus those extracted by acid or alkali, expressed as the mean ± standard deviation (*n* = 3). ^a, b, c, d^ Different letters within the same line indicate differences between in vitro digestion of a compound at significance level *p* < 0.001. Soybean meal aqueous extract was submitted to in vitro gastrointestinal digestion, simulating the oral, gastric and intestinal phases based on human physiology. PD, Pre-digestion; OD, oral digestion; GD, gastric digestion; ID, intestinal digestion.

**Table 3 molecules-24-00074-t003:** Minimal inhibitory concentration of the aqueous extract from soybean meal.

Microorganisms	Soybean Meal Aqueous Extract Concentration (mg/mL)	Inhibition (%)
**Gram-positive**		
*Listeria innocua* ATCC 33090	150	21 ± 2.1
*coagulase*-negative *Staphylococcus saprophyticus* KT955005 *	75	100 ± 0.0
*Staphylococcus aureus* ATCC 14458	120	100 ± 0.0
**Gram-negative**		
*Acinetobacter* genomospecies 3 **	12.5	100 ± 0.0
*Aeromonas hydrophila* FDA110-36	90	100 ± 0.0
*A. hydrophila* ATCC 7966	50	100 ± 0.0
*Escherichia coli* DH5 alfa	35	100 ± 0.0
*E. coli* ATCC 43895	100	100 ± 0.0
*Salmonella enterica* ATCC 12325	75	100 ± 0.0
*S. enterica* ATCC 29934	75	100 ± 0.0
*Vibrio parahemolyticus* ATCC 17802	100	100 ± 0.0
*Pseudomonas fluorescens* ATCC 13525	150	100 ± 0.0

Growth inhibition was determined using different concentrations of the aqueous extract tested on 10^6^ cells of each strains, 12.5 to 150 mg/mL at 37 °C for 18 h. Next, cells were serially diluted (1/10), plated, incubated at 37 °C for 18 h and colony-forming units were counted. Values expressed as the mean ± standard deviation (*n* = 3). * Strain isolated from *Minas* frescal cheese [18]; ** strain isolated from oil refinery sludge [19].

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
