# Peer review of "Recovery of Antimicrobials and Bioaccessible Isoflavones and Phenolics from Soybean (Glycine max) Meal by Aqueous Extraction"

_molecules, 2018, doi:10.3390/molecules24010074_

Round 1

Reviewer 1 Report

I don't recommend the paper for publication in the current status. Some major issues are listed below.

1.      Some notes should be clear. For examples,

a)      Spelling and grammar should be checked

On page 1 line 20

Genistin, daidzin, glycitinand malonylgenistin Genistin, daidzin, glycitin and malonylgenistin

b)      Introduction should be improved in the aspect of water extraction of phytochemicals.

c)       Methods need better explanation

On page 14 line 439

4.5. Evaluation of the antimicrobial activity needs better explanation, controls didn't mentioned at all or standard antibiotic..

Were strains incubated with soybean meal aqueous extracts for 18h, then diluted 1/10 and re-incubated for 18h? Method reference?

Were strains incubated with soybean meal aqueous extracts from 0.0125 to 0.15 mg/ml? How are then concentrations in Table 3. from 35-150 (mg/mL)? Are concentrations maybe in micrograms per mL?

The percentage of growth inhibition is usually determined by the same extract concentrations, not vice versa. Paragraph below Table 3. describes otherwise.

             Growth inhibition was determined using different concentrations of the aqueous extract tested on cells of each strains, 12.5 to 150 mg/mL at 37ºC for 18 h. Next, cells were serially diluted (1/10), plated, incubated at 37°C for 18 h and colony-forming units were counted. Values expressed as the mean ± standard deviation (n = 3).*Strain isolated from Minas frescal cheese

·         ...were grown in NB (nutrient 445 broth) (Himedia, Mumbai, IND) in the presence of crescent concentrations of soybean meal aqueous extract, from 0.0125 to 0.15 mg/mL at 37ºC for 18 h. Then, cells were serially diluted 1/10 in 0.85 % of 447 NaCL, 0.2% Tween 80, plated on solid LB, NB or BHI, and after incubation at 37°C for 18 h the colony forming units were counted. The procedure was performed in triplicate.

On page 15 line 473

             4.6.4. Oxygen radical antioxidant capacity (ORAC)

ORAC assay was performed according to Zulueta et al. [42] with modifications every method modification should be explained.

On page 15 line 515

105cells/mL 105cells/mL

One hundred microliters containing healthy mouse bone marrow (BM) cells (5.0 x 105cells/mL) or fibroblast (1.5 x 105 cells/mL) L929 or human glioblastoma U-87 MG cellswere cultured in 96-well microplates for 24 h at 37°C in 5% CO2 humidified atmosphere

d)     The results were not elucidated in the discussion

Antimicrobial acitivity evaluation of the aqueous extract from soybean meal needs standard reference antbiotic evaluation in the same conditions.

On page 6 line 180 The sentence

 „Soybean meal aqueous extract showed a great antimicrobial activity against both Gram-positive and Gram-negative bacteria (Table 3). „ is not supported by results without appropriate units and reference antibiotic.

e)      Conclusion

Objective conclusions should be based on arranged results

I recommend this paper should be polished up focusing on:

(1) detailed explanation on comments,

 (2) objective conclusions based on results and is resubmitted

Author Response

We believe that we have fully addressed and understood all of Reviewer 1 concerns and comments.

The title was modified (as requested by reviewer 2).

07 new references were included.

Figure 1 was repositioned as Figure S1.

The abstract and discussion of the manuscript were rephrased according to the reviewer's suggestions, following the order of the topic results. The results were discussed in the following order: chemical composition of aqueous extract before and after digestion, antioxidant activity, antimicrobial activity, and cytotoxicity. The same order was adopted in Abstract, Results and Discussion sections. The whole text was revised to improve understanding (as requested by reviewer 2).

All modifications were highlighted in yellow.

After modifications, the manuscript was revised by a specialized editing company in order to improve English grammar and syntax.

The modifications have increased the overall impact of the manuscript. We would like to thank the editor/reviewer for his/her insights and thoughtful critique of our manuscript.

Reviewer 1 comments precede our responses.

Some notes should be clear. For examples,

a)   Spelling and grammar should be checked on page 1 line 20

Answer: A language revision by a specialized company was performed in order to improve English grammar and syntax.

Abstract (page 1, line 20) Genistin, daidzin, glycitinand malonylgenistin → Genistin, daidzin, glycitin and malonylgenistin

Answer: The sentence in page 1, line 20 was modified according to the reviewer's correction.

b)     Introduction should be improved in the aspect of water extraction of phytochemicals.

Answer: The introduction has been improved concerning water extraction of soybean meal and new references were added (page 2, line 47).

Added references:

[5] Xiao, M.; Ye, J.; Tang, X.; Huang, Y. Determination of soybean isoflavones in soybean meal and fermented soybean meal by micellar electrokinetic capillary chromatography (MECC). Food Chem. 2011, 126, 1488-1492, doi: https://doi.org/10.1016/j.foodchem.2010.11.168

[6] Flachowsky, G.; Hünerberg, M.; Meyer, U.; Kammerer, D.R.; Carle, R.; Goerke, M.; Eklund, M. Isoflavone concentration of soybean meal from various origins and transfer of isoflavones into milk of dairy cows. J. Verbr. Lebensm. 2011, 6, 449-456, doi: 10.1007/s00003-011-0702-7.

Methods need better explanation

On page 14 line 439

4.5. Evaluation of the antimicrobial activity→needs better explanation, controls didn`t mentioned at all or standard antibiotic.. Were strains incubated with soybean meal aqueous extracts for 18h, then diluted 1/10 and re-incubated for 18h? Method reference?

Answer: Section 4.5 was modified to better explain the antimicrobial activity and negative control. The reference for this methodology was included (page 14, lines 467).

Unfortunately, at this stage of the study, we cannot compare antimicrobial activities using an isolated control (antibiotic or phenolic compound) because the extract contains a pool of peptides and phenolics compounds, which might produce a synergistic antimicrobial effect that could not be compared to the control. However, we would like to thank the reviewer for the suggestion, which will be very useful for the next steps of future studies.

Were strains incubated with soybean meal aqueous extracts from 0.0125 to 0.15 mg/ml? How are then concentrations in Table 3. from 35-150 (mg/mL)? Are concentrations maybe in micrograms per mL?

Answer: The concentration was mistyped, and is it now corrected (page 14, line 474). The correct concentration ranges from 12.5 to 150 mg/ml, matching the data displayed in Table 3 (page 7).

One new reference, which refers to the origin of Acinetobacter genomospecies 3, was included in Table 3 (page 7).     

Reference:

[60] Pinhati, F. R., Del Aguila, E. M., Tôrres, A. P. R., Sousa, M. P. D., Santiago, V. M. J., Silva, J. T., & Paschoalin, V. M. F. Evaluation of the efficiency of deterioration of aromatic hydrocarbons by bacteria from wastewater treatment plant of oil refinery. Quim. Nova. 2014, 37, 1269-1274, doi: http://dx.doi.org/10.5935/0100-4042.20140221.

The percentage of growth inhibition is usually determined by the same extract concentrations, not vice versa. Paragraph below Table 3. describes otherwise.

Growth inhibition was determined using different concentrations of the aqueous extract tested on cells of each strains, 12.5 to 150 mg/mL at 37ºC for 18 h. Next, cells were serially diluted (1/10), plated, incubated at 37°C for 18 h and colony-forming units were counted. Values expressed as the mean ± standard deviation (n = 3).*Strain isolated from Minas frescal cheese

...were grown in NB (nutrient 445 broth) (Himedia, Mumbai, IND) in the presence of crescent concentrations of soybean meal aqueous extract, from 0.0125 to 0.15 mg/mL at 37ºC for 18 h. Then, cells were serially diluted 1/10 in 0.85 % of 447 NaCL, 0.2% Tween 80, plated on solid LB, NB or BHI, and after incubation at 37°C for 18 h the colony forming units were counted. The procedure was performed in triplicate.

Answer: The text was modified to make it clear that soybean meal aqueous extract concentrations presented in Table 3 correspond to the MIC (minimal inhibitory concentration) for each bacterium, except for L. innocua. Because of this, the soybean meal aqueous extract concentrations are not the same (page 14, lines 477 and page 7).

On page 15 line 473

4.6.4. Oxygen radical antioxidant capacity (ORAC)

ORAC assay was performed according to Zulueta et al. [42] with modifications→ every method modification should be explained.

Answer: The ORAC method was rewritten and all modifications to the method were explained in the text (page 15, line 506).

On page 15 line 515

105cells/mL →105cells/mL

One hundred microliters containing healthy mouse bone marrow (BM) cells (5.0 x 105cells/mL) or fibroblast (1.5 x 105 cells/mL) L929 or human glioblastoma U-87 MG cells were cultured in 96-well microplates for 24 h at 37°C in 5% CO2 humidified atmosphere

Answer: 105cells/mL was corrected in the text (page 16, lines 563 and 564).

d)    The results were not elucidated in the discussion

Antimicrobial activity evaluation of the aqueous extract from soybean meal needs standard reference antibiotic evaluation in the same conditions.

Answer: Unfortunately, at this stage of the study, we cannot compare antimicrobial activities using an isolated control (antibiotic or phenolic compound) because the extract contains a pool of peptides and phenolics compounds, which might produce a synergistic antimicrobial effect that could not be compared to the control. However, we would like to thank the reviewer for the suggestion, which will be very useful for the next steps of future studies.

On page 6 line 180 The sentence

 „Soybean meal aqueous extract showed a great antimicrobial activity against both Gram-positive and Gram-negative bacteria (Table 3). „ is not supported by results without appropriate units and reference antibiotic.

Answer: The reviewer is correct. Since a reference was not used, the text was modified to avoid misinterpretation (page 7, line 159).

e) Conclusion

Objective conclusions should be based on arranged results

I recommend this paper should be polished up focusing on:

(1) detailed explanation on comments,

(2) objective conclusions based on results and is resubmitted

Answer: The conclusion was rewritten with objective conclusions based on results to meet the reviewer's suggestions (page 16, line 584).

Reviewer 2 Report

Please see an attachment file.

Author Response

We believe that we have fully addressed and understood all of Reviewer 2 concerns and comments.

The title was modified (as requested by reviewer 2).

07 new references were included.

Figure 1 was repositioned as Figure S1.

 The abstract and discussion of the manuscript were rephrased according to the reviewer's suggestions, following the order of the topic results. The results were discussed in the following order: chemical composition of aqueous extract before and after digestion, antioxidant activity, antimicrobial activity, and cytotoxicity. The same order was adopted in Abstract, Results and Discussion sections. The whole text was revised to improve understanding (as requested by reviewer 2).

All modifications were highlighted in yellow.

After modifications, the manuscript was revised by a specialized editing company in order to improve English grammar and syntax.

The modifications have increased the overall impact of the manuscript. We would like to thank the editor/reviewer for his/her insights and thoughtful critique of our manuscript.

Reviewer 2 comments precede our responses.

 This manuscript describes polyphenols’ analysis, antimicrobial activity, antioxidant activity before and after digestion test, and cytotoxicity in aqueous extract from soybean meal. It contains some new and valuable findings, but the title and text are not clear and concise. Also, there are so many correctable parts. Please revise and reconstruct it to be easy to read.

Overall

1. The purpose of the present study is obscure. Why do you compile the data for the health-promoting compounds in soybean meal? Are the samples in this study meal or by-products? Have the meal (or by-products) been utilized effectively or unutilized in your country? You should contain ‘The Answer’ in Introduction Section.

Answer: Soybean meal is a secondary product (or by-product) generated after oil extraction from soybean grains. The secondary product is not discarded but widely used for animal feed. A new sentence was included to make this clear (page 2, lines 47-52). Data for the health promoting compounds in soybean meal were not compiled. To avoid misinterpretation, this was excluded (line 77).

2. The figures and tables should be embed in the text near the point where they appear.

Answer: All figures and tables were now embedded in the text as suggested by the reviewer.

3. I do not think that this paper is well written and easy to read. For example, in Abstract section, the authors describes the contents of this research in the order of the chemical composition of aqueous extract (from the meals), the changes in chemical composition after digestion, cytotoxicity of the aqueous extract, antioxidant activity of the aqueous extract before and after digestion, and antimicrobial activity. On the other hand, in Results section, the contents appear in the order of the chemical composition of aqueous extract before and after digestion, antioxidant activity, antimicrobial activity, and cytotoxicity. In Discussion section, they appear in the order of the antimicrobial activity, the chemical composition and antioxidant activity of aqueous extract before and after digestion, and cytotoxicity.

Answer: The reviewer is right. The abstract and discussion of the manuscript were rephrased according to the reviewer's suggestions, following the order of the topic results. The results were discussed in the following order: chemical composition of aqueous extract before and after digestion, antioxidant activity, antimicrobial activity, and cytotoxicity. The same order was adopted in Abstract, Results and Discussion sections. The whole text was revised to improve understanding.

Specific comment:

#Title: Please reconsider. I do not think that this title is based on all sections.

Answer: The title was modified to meet the reviewer's suggestion.

#Abstract: 5-CQA did not appear in the text.

Answer: 5-CQA was replaced by its full name 5-caffeoylquinic chlorogenic acid in the abstract section as required (line 22).

#Overall: hydroxibenzoic acid --->hydroxybenzoic acid (4- or p-hydroxybenzoic acid)

Answer: hydroxybenzoic acid was replaced by 4-hydroxybenzoic acid (Lines 96, 98, 236, 243, 264, 279).

#L113: galic acid ---> gallic acid

Answer: galic acid was replaced by gallic acid (line 107).

#Figure 1 or L479-488: Please describe concentrations of synthetic antioxidants (BHA, BHT, and α-TOC). Please add control data (without antioxidant) in Figure 1. (I suppose the paper would be better, if the authors move this figure to Supplementary Materials. Because this data is of minor importance, and it is difficult to combine data in Figure 1 and Figure 2).

Answer: The concentrations of synthetic antioxidants (BHA, BHT, α-tocopherol and samples) was added to the text (page 15, line 531). The control (data obtained without antioxidant) was added to Figure S1. Figure 1 was moved to supplementary materials as Figure S1 as suggested by the reviewer.

#Table 3 and L156-168: Is the unit (mg/mL) correct? ug/mL? In L446, the concentrations are from 0.0125 to 0.15 mg/mL.

Answer: The reviewer is correct. In the antimicrobial assays, the soybean meal aqueous extract was used in concentrations varying from 12.5 to 150 mg/mL. This information was corrected in line 474. Cytotoxicity studies, which are described in line 160, were independently performed and the extract was used in concentrations ranging from 125 mg/mL to 0.97 mg/mL.

#Table 1 and 2: ID, intestinal digestion in Table 1 and 2, but DD, duodenal digestion in the text (L492). Please unify.

Answer: Initials were unified as suggested (page 16, line 541 and 543).

#Table 1: Please confirm the value of SD. All values are 0.0.

Answer: SD values are not zero but became to zero when approximations were applied. Considering the reviewer suggestion, the SD values were included without approximation and with two/three decimal separators.

#Table 2: 4-hydroxyphenilacteic acid ---> 4-hydroxyphenylactetic acid

Please move ‘Hydroxybenzoic acid’ data to Hydroxybenzoics column.

Please confirm the data of p-coumaric acid (1.0±0.01).

Answer: 4-hydroxyphenylactetic acid was corrected and Hydroxybenzoic acid was moved to Hydroxybenzoics column as suggested by the reviewer.

The phenolic compoundsanalysis used in the present study was based on the methodology proposed by Inada et al. (2015) using two methods for extracting these compounds from the soybean meal aqueous extract. Soluble and insoluble phenolic compounds were extracted. Aliquots were drawn from each extraction and analyzed by HPLC. The results of these extractions are described in Table 2 in the pre-digestion (PD) column, i.e, the results of extracted phenolic compounds (underwent two extraction procedures) from the soybean meal aqueous extract.

However, we performed the analysis of the phenolic compounds in the soybean meal aqueous extract after in vitro human simulated gastrointestinal digestion. We performed this in vitro digestion because the digestive process in the human body is a complex phenomenon involving ingestion, hydrolysis, absorption, secretion and transit, making it difficult to study such an event in vivo and to determine the digestibility of food products, since several interactions between different processes occurring simultaneously are noted.

After each stage of the digestive process (oral, gastric and intestinal), an aliquot was removed (from each stage) for the analysis of phenolic compounds. However, we did not perform the aforementioned extracting phenolic compound methods in these aliquots. Our objective was to observe bioaccessibility, that is, the fraction or content of phenolic compounds released from the alimentary matrix (soybean meal aqueous extract), in the gastrointestinal tract during the in vitro simulated gastrointestinal digestion that would become available for human absorption in intestinal mucosa. During the oral phase, we observed that phenolic compounds percentage of release was reduced for all phenolic compounds when compared to PD. However, some phenolic compounds such as caffeic acid, gallic acid, vanillic acid and rutin increased significantly after the gastric and intestinal phase. Other phenolic compounds such as 4-hydroxybenzoic acid, 5-caffeoylquinic chlorogenic acid and hesperidin increased significantly after the gastric phase but decreased after the intestinal phase. These results are in agreement with previous studies that concluded that the stability and antioxidant activity of phenolic compounds depend on the physicochemical conditions in the digestive tract segments, such as pH, temperature and enzyme activities, as well as the nature of the food matrix (Bouayed et al., 2011; Baião et al., 2017b). The racemization of phenolic compounds occurs, altering and rendering more reactive compounds than in the beginning of the digestive process. On the other hand, the alkaline pH during intestinal digestion may reduce racemization (forming fewer reactive compounds) and degrade phenolic compounds, leading to loss of activity and bioaccessibility (Baião et al., 2017b; Ryan et al., 2010). Furthermore, some phenolic compounds are sensitive to the alkaline pH in duodenal fluid, and may be modified into distinct structures, probably their non-detectable aglycone forms, facilitating absorption through epithelial cells in the duodenal mucosa (Manach et al., 2005) (page 10, lines 266).

We have revised the entire section text in order to improve understanding.

References:

Inada, K. O. P.; Oliveira, A. A.; Revorêdo, T. B.; Martins, A. B. N.; Lacerda, E. C. Q.; Freire, A. S.; Braz, B. F.; Santelli, R. E.; Torres, A. G.; Perrone, D.; Monteiro, M. C. Screening of the chemical composition and occurring antioxidants in jabuticaba (Myrciariajaboticaba) and jussara (Euterpe edulis) fruits and their fractions. J. Funct. Foods2015, 17, 422-433, doi: https://doi.org/10.1016/j.jff.2015.06.002.

Bouayed, J.; Hoffmann, L.; Bohn, T. Total phenolics, flavonoids, anthocyanins and antioxidant activity following simulated gastro-intestinal digestion and dialysis of apple varieties: bioaccessibility and potential uptake. Food Chem. 2011, 128, 14-21. doi: 10.1016/j.foodchem.2011.02.052.

Baião, D.S.; Freitas, C.S.; Gomes, L.P.; da Silva, D.; Correa, A.C.N.T.F.; Pereira, P.R.; Del Aguila, E.M.; Paschoalin, V.M.F. Polyphenols from root, tubercles and grains cropped in Brazil: chemical and nutritional characterization and their effects on human health and diseases. Nutrients. 2017b, 9, 1-29. doi: 10.3390/nu9091044.

Ryan, L.; Prescott, S.L. Stability of the antioxidant capacity of twenty-five commercially available fruit juices subjected to an in vitro digestion. Intl. J. Food Sci. Technol. 2010, 45, 1191-1197. doi: https://doi.org/10.1111/j.1365-2621.2010.02254.x.

Manach, C.; Williamson, G.; Morand, C.; Scalbert, A.; Rémésy, C. Bioavailability and bioefficacy of polyphenols in humans. I. Review of 97 bioavailability studies. Am. J. Clin. Nutr. 2005, 81, S230–S242. doi: 10.1093/ajcn/81.1.230S.

#L222: What did show 3-fold the contents described in soy protein isolate? Total isoflavons in aqueous extract?

Answer: the reviewer is correct, 3-fold refers to total isoflavone content. The text was modified to make this clear (page 5, line 117 and page 9, line 213).

#L239: cafeic acid ---> caffeic acid. o-cummaric acid? In a ref#4, the identified phenolic acids are vanillic, syringic, coumaric, ferulic, p-hydroxybenzoic and sinapic acids. Caffeic acid is not identified in ref#4.

Answer: cafeic acid was replaced by caffeic acid (page 9, line 240 and 243) and o-cummaric acid was excluded (line 245)

#L390: Why did you heat-treat the supernatant after extraction? Inactivation of enzyme?

Answer: The sentence was modified to make this clear, including the explanation for the heat treatment step (lines 415).

#L392: LC-DAD-FL. Please explain FL.

Answer: There is a “D” missing in FL. The correct initial is FLD, which means Fluorescence Detector (page 13, lines 418, 418 and 422).

A photodiode array detector is abbreviated as DAD (diode arrangement detector, L392) or PDA (L429). Please unify.

Answer: Initials were unified as required. PDA was replaced by DAD (lines 419, 422, 428, 431, 456 and 543).

#L407: Please confirm the concentration (0.1 – 1.0 ppm) of standards to make calibration curves. The dynamic range is too narrow to quantify isoflavones in the aqueous extracts treated with or without digestion test.

Answer: the concentration (0.1 - 1.0 ppm) of the standards used in rhe calibration curves was incorrect. The correct value is 0.1 - 10.0 ppm. This sentence was corrected in the text (page 13, line 434)

#L489: The analytical data of flavones in the human feces are very valuable. Please show appreciate experimental protocol (daily intake of soybean meal, times, days, and so on).

Answer: The ex vivo colonic fermentation was performed according to the methodologies described by Hu et al. (2004) and Mosele et al. (2015) with modifications. As explained in the text, volunteers did not intake soybean meal. They only donated feces (page 16, line 549). In order to donate the feces, volunteers had to meet inclusion criteria (page 16, line 549).

 The feces were homogenized in a nutrient-rich medium (0.5 g in 10 ml) described by McDonald et al. (2013) and 5.0 mL of this mixture was added to the in vitro simulated gastrointestinal digested material at the end of the entire simulated digestion process, incubated at 37°C and 50 rpm for 48 h.

References:

Hu, J.; Zheng, Y. L.; Hyde, W.; Hendrich, S.; Murphy, P. A. Human fecal metabolism of soyasaponin I. J. Agric. Food Chem.2004. 52, 2689–2696, doi: 10.1021/jf035290s.

Mosele, J. I.; Macià, A.; Romero, M. P.; Motilva, M. J.; Rubió, L. Application of in vitro gastrointestinal digestion and colonic fermentation models to pomegranate products (juice, pulp and peel extract) to study the stability and catabolism of phenolic compounds. J. Func. Foods. 2015, 14, 529–540, doi: https://doi.org/10.1016/j.jff.2015.02.026.

McDonald, J. A.; Schroeter, K.; Fuentes, S.; Heikamp-deJong, I.; Khursigara, C. M.; de Vos, W. M.; Allen-Vercoe, E. Evaluation of microbial community reproducibility, stability and composition in a human distal gut chemostat model. J. Microbiol. Methods. 2013, 95, 167-174, doi: 10.1016/j.mimet.2013.08.008.

Reviewer 3 Report

The article is very well written. I have only three technical comments.

Line 26: "in vitro" please write in italics.

Please insert full stops after table titles.

Lines 59 and 63: Please add one more reference (the same in both aforementioned lines) and enter it in the reference list.

Budryn, G.; Gałązka-Czarnecka, I.; Brzozowska, E.; Grzelczyk, J.; Mostowski, R.; Żyżelewicz, D.; Cerón-Carrasco, J. P.; Pérez-Sánchez, H. Evaluation of estrogenic activity of red clover (Trifolium pratense L.) sprouts cultivated under different conditions by content of isoflavones, calorimetric study and molecular modelling. Food Chem. 2018, 245, 324-336, (doi.org/10.1016/j.foodchem.2017.10.100).

Author Response

We believe that we have fully addressed and understood all of Reviewer 3 concerns and comments.

The title was modified (as requested by reviewer 2).

07 new references were included.

Figure 1 was repositioned as Figure S1.

The abstract and discussion of the manuscript were rephrased according to the reviewer's suggestions, following the order of the topic results. The results were discussed in the following order: chemical composition of aqueous extract before and after digestion, antioxidant activity, antimicrobial activity, and cytotoxicity. The same order was adopted in Abstract, Results and Discussion sections. The whole text was revised to improve understanding (as requested by reviewer 2).

All modifications were highlighted in yellow.

After modifications, the manuscript was revised by a specialized editing company in order to improve English grammar and syntax.

The modifications have increased the overall impact of the manuscript. We would like to thank the editor/reviewer for his/her insights and thoughtful critique of our manuscript.

Reviewer 3 comments precede our responses.

               1.     Line 26: “in vitro” please write in italics.

Answers: the sentence in vitro was rewritten (page 1, line 24).

2.     Please insert full stops after table titles.

Answers: full stops were added after all table titles.

3.     Lines 59 and 63: Please add one more reference (the same in both aforementioned lines) and enter it in the reference list.

Answer: New references were added in lines 67 and 69 to meet reviewer suggestions. Reference numbers were rearranged throughout the text and in the reference list.

References added:

 [9] Bursać, M.; Krstonošić, M. A.; Miladinović, J.; Malenčić, D.; Gvozdenović, L.; Cvejić, J. H. Isoflavone Composition, Total Phenolic Content and Antioxidant Capacity of Soybeans with Colored Seed Coat. Nat. Prod. Comm. 2017, 12, 527-532, doi: 10.1089/jmf.2009.0050.

[10] Budryn, G.; Gałązka-Czarnecka, I.; Brzozowska, E.; Grzelczyk, J.; Mostowski, R.; Żyżelewicz, D.; Cerón-Carrasco, J. P.; Pérez-Sánchez, H. Evaluation of estrogenic activity of red clover (Trifolium pratense L.) sprouts cultivated under different conditions by content of isoflavones, calorimetric study and molecular modelling. Food Chem. 2018, 245, 324-336, doi: 10.1016/j.foodchem.2017.10.100. 

[11] Lee, J. H.; Lee, B. W.; Kim. B.; Kim, H. T.; Ko, J. M.; Baek, I. Y.; Seo, W. T.; Kang, Y. M.; Cho, K. M. Changes in Phenolic Compounds (Isoflavones and Phenolic acids) and Antioxidant Properties in High-Protein Soybean (Glycine max L. cv. Saedanbaek) for Different Roasting Conditions. J. Korean Soc. Appl. Biol. Chem. 2013, 56, 605-612, doi: 10.1007/s13765-013-3048-2.

Round 2

Reviewer 1 Report

It is not clear why did you highlight entire susections when you have changed only a few words, you should only highlight the changes (Abstract, The main objective of the present study, Discussion,..), so it is difficult to keep track of changes. In Subsubsection 2.2.2. Antimicrobial activity you didn't change anything, but the entire is highlighted. The main reason is?

Soybean meal aqueous extracts concentration from 12.5 to 150 mg/ml are too high for a statement that they have good antibacterial properties, and the author's response for subsection 4.5. Evaluation of the antimicrobial activity for the lack of standard antibiotic is not adequate but lack of it is acceptable. 

Reviewer 2 Report

I think appropriate modifications were made in revised version.